# RPIPLM: Prediction of ncRNA-protein interaction by post-training a dual-tower pretrained biological model with supervised contrastive learning

**Yiwei Liu**[1], **Ting Bao**[2], **Peng Yin**[1,3]*, **Shumin Wang**[4], **Yanbin Wang**[5]*

**1** Defence Industry Secrecy Examination and Certification Center, Beijing, China, **2** National Key Laboratory of Science and Technology on Information System Security, Beijing, China, **3** School of Cyber Security, University of Chinese Academy of Sciences, Beijing, China, **4** Department of Obstetrics and Gynecology, The Affiliated Hospital of Inner Mongolia Medical University, Huhhot, China, **5** Shenzhen MSU-BIT University, Shenzhen, China

* wangyanbin15@mails.ucas.ac.cn (YW); yinpeng@iie.ac.cn (PY)

**Data availability statement:** All data files are available from the public database mentioned in

## Abstract

The field of biological research has been profoundly impacted by the emergence of biological pre-trained models, which have resulted in remarkable advancements in life sciences and medicine. However, the current landscape of biological pre-trained language models suffers from a shortcoming, i.e., their inability to grasp the intricacies of molecular interactions, such as ncRNA-protein interactions. It is in this context that our paper introduces a two-tower computational framework, termed RPIPLM, which brings forth a new paradigm for the prediction of ncRNA-protein interactions. The core of RPIPLM lies in its harnessing of the pre-trained RNA language model and protein language model to process ncRNA and protein sequences, thereby enabling the transfer of the general knowledge gained from self-supervised learning of vast data to ncRNA-protein interaction tasks. Additionally, to learn the intricate interaction patterns between RNA and protein embeddings across diverse scales, we employ a fusion of scaled dot-product self-attention mechanism and Multi-scale convolution operations on the output of the dual-tower architecture, effectively capturing both global and local information. Furthermore, we introduce supervised contrastive learning into the training of RPIPLM, enabling the model to effectively capture discriminative information by distinguishing between interacting and non-interacting samples in the learned representations. Through extensive experiments and an interpretability study, we demonstrate the effectiveness of RPIPLM and its superiority over other methods, establishing new state-of-the-art performance. RPIPLM is a powerful and scalable computational framework that holds the potential to unlock enormous insights from vast biological data, thereby accelerating the discovery of molecular interactions.

this manuscript: 1. Data NPInter2.0 is available from https://doi.org/10.1093/nar/gkt1057; 2. NPI7317 is available from https://doi.org/10.1093/database/baw057; 3. RPI2241, RPI369, RPI1807 are available from https://doi.org/10.1093/nar/gkq1108.

**Funding:** The author(s) received no specific funding for this work.

**Competing interests:** The authors have declared that no competing interests exist.

## Introduction

Non-coding RNAs (ncRNAs), which do not encode proteins, were previously regarded as "junk" in the genome by some researchers. However, recent years have witnessed the discovery of various functionally significant ncRNA types [1–3], including microRNA, small interfering RNA (siRNA), Piwi-interacting RNA (piRNA), small nuclear RNA (snRNA), and long non-coding RNAs (lncRNAs) such as Xist and HOTAIR. These ncRNAs play pivotal roles in a range of biological processes, including gene expression regulation, chromatin modification, and disease development, among others. As such, they have attracted considerable attention in the scientific community and have become the focus of numerous studies aimed at elucidating their functions and mechanisms. The discovery of these functionally important ncRNAs has revolutionized our understanding of the genome and opened up new avenues for research in the field of molecular biology.

Functional non-coding RNAs not only play vital roles in transcriptional and post-transcriptional regulation but also serve as essential regulators of gene expression at the epigenetic level [4,5]. As such, the accurate prediction of ncRNA-protein interactions is critical. Consequently, the prediction of ncRNA-protein interactions has become a significant research topic in the field of molecular biology, with numerous computational methods proposed for this purpose. By accurately predicting ncRNA-protein interactions, we can gain a deeper understanding of the underlying molecular mechanisms and develop new therapeutic strategies for various diseases.

Experimental techniques like PAR-CLIP and RIP-Chip have facilitated the discovery of ncRNA-protein interactions (RPIs), albeit with considerable expenses, labor, and time constraints. Alternatively, computational methods provide a cost-effective and efficient means to predict novel RPIs [6]. Leveraging matrix manipulation, classical machine learning, and advanced neural networks, these computational approaches analyze vast biological datasets and predict RPIs accurately. By employing computational methods, researchers can surmount the limitations of experimental techniques, uncovering new RPIs, enhancing our understanding of molecular mechanisms, and potentially developing novel therapeutic strategies. Notable computational methods include lncPro [7], RPISeq [8], RPI-Pred [9], LPIHN [10], and LPBNI [11], each employing distinct strategies such as matrix multiplication, k-mer encoding, support vector machine (SVM) training, and network-based algorithms to predict RPIs.

Deep learning has emerged as a promising approach for predicting non-coding RNA-protein interactions (RPIs) [12,13]. IPMine [14] employs stacked autoencoders to extract embedded features of ncRNA and protein sequences, and uses integrated strategies to build robust RPI predictors. LncADeep [15] utilizes a deep belief network to predict RPIs using sequence and structure information of RNA and protein. LPI-BLS [16] infers lncRNA-protein interactions by training multiple broad learning models and integrating the outputs of all models using logistic regression. DM-RPIs [17] incorporate convolutional neural networks and extreme learning machine classifiers to predict ncRNA-protein interactions. RPI-CNNBLAST [18] utilizes three convolutional neural networks to learn abstract features of RNA sequence, RNA structure and protein sequence information, and fuses them with a BLAST network to train an RPI predictor. RPI-SE [19] is an ensemble learning model that uses RNA sequence information and protein evolution information to predict the interaction between ncRNA and protein. LPI-Pred [12] trains two biological sequence word representation learning models, RNA2vec and Pro2vec, to generate distribution representations of RNA and proteins, and trains a random forest classifier using high semantic features screened by Gini information impurity to predict lncRNA-protein interactions. EDLMFC [20]

uses a convolutional neural network to automatically fuse multi-scale RNA and protein features, and a bidirectional long-short-term memory network (BiLSTM) to capture long-range dependencies between CNN features.

Graph representation learning has been widely used in recent years to predict potential relationships between entities in various biological networks, including RNA and protein networks [21–23]. Wekesa et al. [24] proposed an LSTM with graph attention to learn lncRNA and protein embedding representations from known lncRNA-protein interaction networks. NPI-DGCN [22], a dual graph convolutional network-based RPI predictor, utilizes two identical GCNs to extract RNA and protein embeddings and maps them to a unified representation space using a learnable filter matrix.

The above methods make valuable contributions to the prediction of RNA-protein interactions. However, these methods are all limited to labeled interaction data, and the available RNA-protein interaction data only cover a small part of the whole interaction network. It is well known that when the training data is very limited, the model usually does not generalize well due to the obvious difference between the training set and the real world. Therefore, it is of great significance to explore new computing paradigms that allow machine learning algorithms to learn general knowledge from massive unlabeled data and transfer the learned knowledge to the learning of labeled data. This can potentially unlock new insights into the underlying mechanisms of gene regulation and facilitate the development of novel therapeutic strategies.

The pre-trained language model (PLM) is a powerful machine learning tool that employs the computationally efficient Transformer architecture and a revolutionary self-supervised learning method to train on vast amounts of unlabeled data. This method has shown impressive performance in various machine learning applications and has established the new paradigm of "pretraining-fine-tuning". In bioinformatics, PLMs such as ProteinBERT [25], MSA Transformer [26], ProtTrans [27], RNABERT [28], AminoBERT [29], DNABERT [30], and others [31–34] have been developed for various applications, but are limited to protein or nucleotide-based molecules. As a result, these models cannot be directly applied to predict molecular interactions such as ncRNA-protein interactions.

In this study, we present RPIPLM, a pioneering computational framework designed to revolutionize the prediction of ncRNA-protein interactions. RPIPLM is a two-tower pre-trained language model that combines RNA and protein pre-trained language models, allowing for the efficient extraction of general features from RNA and protein sequences in parallel. By cleverly leveraging self-supervised learning on massive unlabeled datasets, RPIPLM is capable of transferring knowledge gained from these data sets to the prediction of ncRNA-protein interactions. Building upon the foundation of pre-trained embeddings, RPIPLM employs a fusion of dot-product self-attention mechanism and one-dimensional convolutional operations to capture intricate interaction patterns between RNA and protein embeddings across diverse scales. Additionally, it incorporates supervised contrastive learning to effectively capture discriminative information. RPIPLM represents a significant advancement in the field of RNA-protein interaction prediction, and has the potential to unlock a deeper understanding of these critical biological interactions, opening up new avenues for research in the life sciences and medicine. Our contribution is as follows:

- As far as current research knowledge is concerned, our proposed method stands out as the first of its kind to leverage pre-trained language models in the prediction of biomolecular interactions by ingeniously integrating two distinct classes of biological pre-trained models.

This innovative approach establishes a new computational framework for biomolecular interaction prediction.

- RPIPLM sets a new standard for predicting biomolecular interactions by leveraging the power of pretrained language models, local and global attention, and supervised CL, achieving state-of-the-art performance. Through extensive experiments and an interpretability study, RPIPLM showcases its exceptional ability to comprehend biomolecular interaction patterns.
- We present empirical evidence demonstrating that RPIPLM outperforms information fusion-based approaches for predicting RNA-protein interactions, leveraging the power of unsupervised learning. This finding not only underscores the potential of unsupervised learning techniques in this domain but also highlights the existence of a wealth of intricate information within sequences, which may extend beyond our current understanding.

## The proposed method

In this section, we describe the proposed RIPPLM, a twin-tower model with protein pretrained models and RNA pretrained modelsfor accurate ncRNA-protein interaction prediction. The proposed model is composed of three distinct modules: the two-tower module, feature fusion module, and predictor module, which are elaborately designed to extract and integrate essential features from RNA and protein sequences. The overall architecture of the proposed model is illustrated in Fig 1.

### Two-towers module

The twin-tower module comprises two biologically pre-trained models derived from vast unlabeled protein and RNA sequence data, one each for protein and RNA. Its purpose is to transfer general knowledge learned by the pre-training model from extensive unlabeled data to the task data, thereby obtaining sequence features of RNA and protein with abstract semantic information.

We adopt RNABERT as the RNA branch of our twin-tower model, leveraging its ability to effectively capture complex RNA sequence information. In contrast, for protein feature extraction, we comprehensively evaluate three state-of-the-art encoder models, namely BERT [35], Albert [36], and Electra [37], using the ProtTrans tool [27]. By conducting rigorous evaluations, we aim to identify the best-performing protein pre-trained model to be used in our RPI prediction framework.

**RNABERT.** RNABERT effectively encodes RNA sequences by utilizing a BERT network that is pre-trained on a vast amount of unlabeled non-coding RNAs (ncRNAs). The network architecture of RNABERT is identical to that of BERT, except for the usage of only 6 layers of Transformer encoders. For further information, please refer to [**?** ]. To encode each base (A, C, G, U) in the input RNA sequence, RNABERT utilizes the classic masked language model (MLM) task and structural alignment learning (SAL) for self-supervised training, resulting in a 120-dimensional vector representation for each base.

In the MLM task, 15% of the input RNA sequence is masked, and the model predicts the masked portion using the surrounding bases. The MLM task was trained on 762,370 sequences generated by applying 10 different mask patterns to 76,237 human ncRNA queries. The SAL task, on the other hand, aims to learn the relationship between two RNA sequences and improve base embeddings by aligning RNA structures. RNABERT utilizes the Rfam seed alignment as the reference structural alignment for the SAL task. The SAL mega task utilizes a matrix $\Omega$ to compute the alignment score matrix of two RNA sequences $n$ and $m$, where $E^n$ and $E^m$ are the embedded representations output from the transformer layer for the input of

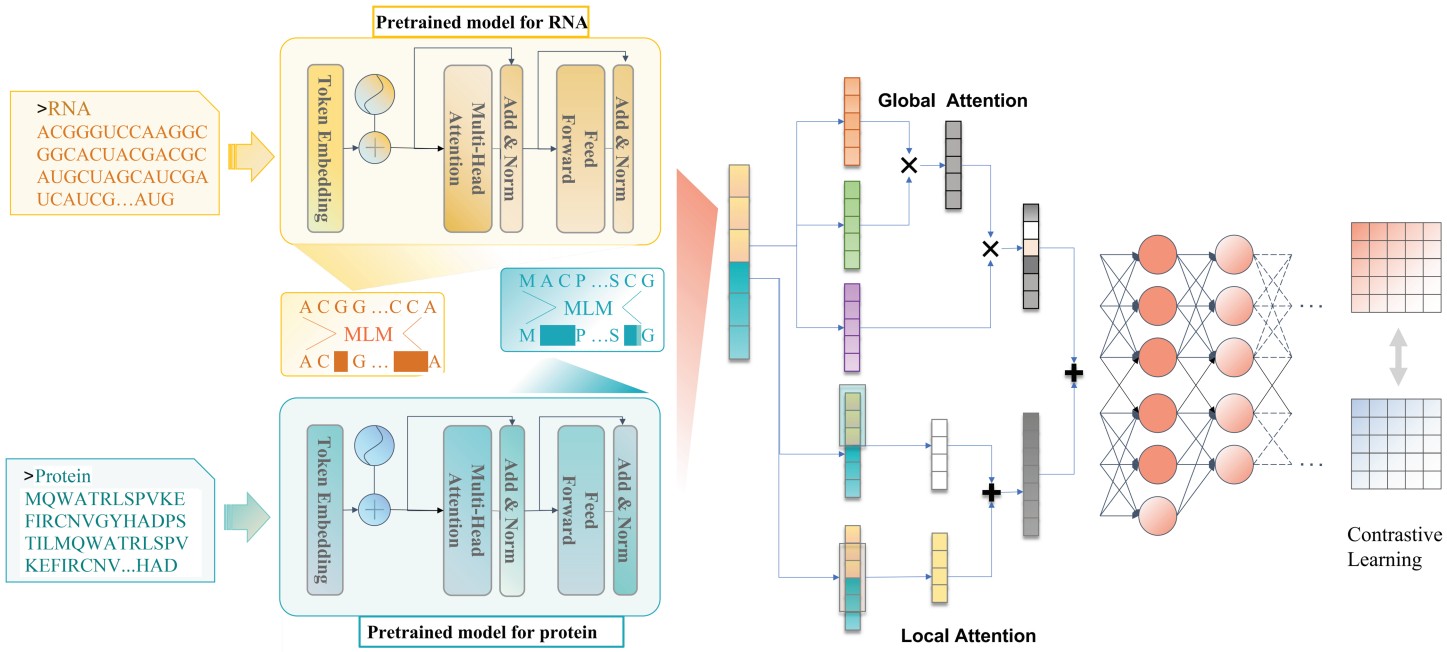

**Fig 1. Architecture diagram of RIPPLM.**

two RNA sequences. Each element $\Omega_{ij}$ is defined to be the normalized inner product between $E_i^n$ and $E_j^m$.

RNABERT applies binary classification learning to assign $\omega_{ij}$ scores, with aligned locations scoring close to 1 and unaligned locations scoring close to 0. For a training set $D$, consisting of triplets $(x, x^p, y)$, where $x$ and $x^p$ are a pair of RNA sequences and $y$ is the corresponding reference alignment between $x$ and $x^p$. RNABERT aims to minimize the following loss function $L$:

$$L = \sum_{(x, x^p, y) \in D} f(x, x^p, \hat{y}) + \delta(y, \hat{y}) - f(x, x^p, y) + \lambda \, \|\omega_2\|, \tag{1}$$

where $f$ is the function that returns the alignment score y between $x$ and $x^p$. $\|\omega_2\|$ is the squared value of the model parameters and $\lambda$ is a parameter that controls the strength of regularization. $\hat{y}$ is the predicted alignment path calculated by the Needleman–Wunsch algorithm to maximize the sum of the alignment score $f(x, x^p, \hat{y})$ and the margin term $\delta(y, \hat{y})$. The margin term $\delta(y, \hat{y})$ defines the difference between the reference alignment and the predicted alignment as follows:

$$\delta(y, \hat{y}) = \epsilon \times num + \zeta \times n\hat{u}m, \tag{2}$$

where $\epsilon$ and $\zeta$ are hyperparameters that control the trade-off between sensitivity and specificity for learning parameters, which are set to 0.05 and 0.1, respectively. *num* demotes the number of positions included in $y$ but not in $\hat{y}$, and *n$\hat{u}$m* demotes the number of positions included in $\hat{y}$ but not in $y$.

**Predicted protein language models.** In the protein branch of our twin-tower model, we evaluate five pre-trained models, including two versions of ProtBERT, a ProtAlbert, and a ProtElectra, which are obtained by training classic language models (BERT, Albert, and Electra) on protein sequences. ProtAlbert and ProtElectra improve on the BERT architecture by

sharing hard parameters between attention layers, allowing for increased attention heads and utilizing an adversarial strategy to improve pre-training task sampling efficiency. The Prot-Trans toolkit is used to leverage these three models. The fifth model, OntoProtein, incorporates protein Gene Ontology (GO) information to enhance protein pre-training, a departure from the previous models that rely solely on protein sequences. Table 1 lists the model hyperparameters.

**ProtBERT.** The ProtBERT model, designed for protein sequence pre-training, is available in two versions: ProtBert-Ref100 and ProtBert-BFD. The former uses UniRef100 as training data, while the latter employs BFD-100 for training. The ProtBERT model has a deeper architecture than the original BERT model, and its training procedure is similar to that in the paper by Lamb et al. [38]. Specifically, ProtBert-Ref100 is trained for 400K steps, while ProtBert-BFD is trained for 1000K steps.

**ProtAlbert.** ProtAlbert is pre-trained on the UniRef100 dataset using the configuration from Albert's official GitHub repository, version: *xxlarge v2*. It increases the global batch size from 4096 to 10752, which allowed us to leverage the entire diversity of the protein universe, given that the vocabulary in proteins is much smaller than that of natural language. ProtAlbert is trained for 3000K.

**ProtElectra.** ProtElectra is trained on the UniRef100 dataset using Electra's official configuration with some modifications. The number of layers is increased to 30, and the *Lamb* optimizer is used to improve training efficiency. The adversarial training strategy is implemented through generators and discriminators with a smaller hidden layer size (25% of the discriminator hidden layer size). ProtElectra is trained for 800K steps.

**OntoProtein.** OntoProtein is a protein OntoProtein is a approach that integrates the gene ontology knowledge graph to augment the representation of proteins. Through the creation of ProteinKG25, a knowledge graph built from the Swiss-Prot database and the public gene ontology library, OntoProtein incorporates over 4 million protein-GO triplets and 100,000 GO-GO triplets. By jointly optimizing the protein sequence MLM self-supervised learning objective and GO knowledge graph embedding learning, OntoProtein enhances protein representation through the introduction of functional information from the GO knowledge in protein pre-training.

In OntoProtein, ProtBERT is used to encode proteins pre-trained on the UniRef100 dataset, while PubMedBERT is employed to encode biological descriptions in gene ontology

**Table 1. Model hyperparameters: This table shows the 6 large-scale pre-trained models involved in our twin tower model. Note: The attention model parameters are not displayed in this parameter table, as they have already been explained in the attention mechanism section.**

| Hyperparameter | RNA branch | Protein branch | | | | | Concatenation Network |
| --- | --- | --- | --- | --- | --- | --- | --- |
| | RNABERT | ProtBert | | ProtAlbert | ProtElectra | Ontoprotein | Network |
| Dataset | RNAcentral | BFD100 | UniRef100 | UniRef100 | UniRef100 | UniRef100+ProteinKG25 | |
| Number of Layers | 6 | 30 | | 12 | 30 | 30 | 2 |
| Hidden Layers Size | 768 | 1024 | | 4096 | 1024 | 1024 | 512 |
| Hidden Layers Intermediate Size | 3072 | 4096 | | 16384 | 4096 | 4096 | |
| Number of Heads | 12 | 16 | | 64 | 16 | 16 | |
| Dropout | 0.1 | 0.0 | | 0.0 | 0.0 | 0.0 | 0.1 |
| Target Length | 120 | 512/2048 | | 512/2048 | 512/1024 | 512/2048 | |
| Pretraining Tasks | MLM(15%) SAL | MLM(15%) | | MLM(15%) | MLM(15%) | MLM+KE | |
| Local Batch Size | 40 | 32/6 | 30/5 | 2172 | 18/7 | 30/5 | |
| Optimizer | AdamW | Lamb | | Lamb | Lamb | **Lamb** | AdaBound |
| Learning Rate | 0.001 | 0.002 | | 0.002 | 0.002 | 0.002 | 0.001 |

entities, which are utilized for knowledge graph embedding learning. In essence, OntoProtein represents an informative approach that enhances sequence modeling (protein sequence pre-training) through the integration of knowledge graphs.

## Concatenation

To parallelize the RNA and protein branches, we attach a connection layer on top of the two pre-trained models and concatenate the output of the two models, specifically the embeddings corresponding to the CLS token, to construct RNA-protein pairs. This allows for efficient knowledge transfer from the pre-trained models to the prediction task. o evaluate the impact of different concatenations of RNA-protein pairs on the RPI prediction task, we consider the following concatenation methods:

- Direct splicing ($r,p$): Concatenating RNA and protein embeddings without any transformation. This method provides a straightforward approach to concatenate RNA and protein features, but may not capture more complex interactions between RNA and protein.
- Modulus of difference ($|r-p|$): Computing the modulus of the difference between the RNA and protein embeddings. This method focuses on capturing the differences between the RNA and protein features, but may not fully capture the interaction patterns between them [39].
- Element-wise product ($r*p$): Computing the element-wise product of RNA and protein embeddings. This method focuses on capturing the interaction patterns between RNA and protein features, but may not fully capture the differences between them .
- Concatenation ($|r-p|, r*p$): Concatenating the modulus of the difference between the RNA and protein embeddings and the element-wise product of RNA and protein embeddings. This method combines the strengths of the previous two methods to capture both the differences and interactions between RNA and protein features.
- Concatenation ($r,p,r*p$): Concatenating RNA and protein embeddings and their element-wise product. This method allows for direct interpretation of RNA and protein features and their interaction patterns.
- Concatenation ($r, p, |r-p|$): Concatenating RNA and protein embeddings and the modulus of their difference. This method focuses on capturing both the differences and similarities between RNA and protein features, but may not fully capture the interaction patterns between them.
- Concatenation ($r, p, |r-p|, r * p$): Concatenating RNA and protein embeddings, the modulus of their difference, and their element-wise product. This method combines the strengths of the previous three methods to capture both the differences, similarities, and interaction patterns between RNA and protein features.

By comparing the performance of these different concatenation methods, we aim to identify the most effective way to construct RNA-protein pairs in RPIPLM for accurate interaction prediction.

## Global and local attention modules

In order to capture both global and local patterns in the concatenated RNA and protein embeddings, we employ a combination of dot-product self-attention mechanism (applied along the sequence dimension) and one-dimensional convolutional operations. Specifically, given the sequence of concatenated token embeddings $X = [x_1, x_2, ..., x_n]$, where $n$ is

the sequence length and each $x_i \in \mathbb{R}^d$ is a feature vector, our attention mechanism computes dependencies across different tokens in the sequence, not across the feature channels.

**Global attention: Scaled dot-product attention.**   The global attention mechanism is based on the scaled dot-product attention mechanism [40]. Given an input sequence of embeddings $X = x_1, x_2, \ldots, x_n$, where $n$ is the sequence length, the scaled dot-product attention computes the attention score for each element in the sequence. The attention score is calculated using the following equation:

$$A_{ij} = \frac{\exp\left(X_i X_j^T / \sqrt{d}\right)}{\sum_{l=1}^{n} \exp\left(X_i X_l^T / \sqrt{d}\right)}, \tag{3}$$

where $X_i$ and $X_j$ are the input embeddings for the $i$-th and $j$-th elements in the input sequence, respectively, and $d$ is the dimension of the input embeddings. The output of the global attention layer is obtained by multiplying the attention scores with the input embeddings:

$$Y_i^{global} = \sum_{j=1}^{n} A_{ij} X_j. \tag{4}$$

**Local attention: Multi-scale convolution.**   To capture local dependencies at different scales and depths, we employ multi-scale, multi-layer one-dimensional (1D) convolution layers. For each scale, we stack multiple 1D convolution layers with a specific kernel size. The intuition behind using multiple kernel sizes and depths is to allow the model to learn patterns with varying lengths and complexities in the concatenated embeddings of RNA and protein sequences.

Given the concatenated embeddings of RNA and protein sequences, $X = x_1, x_2, \ldots, x_n$, we apply $M$ stacks of 1D convolution layers with kernel sizes $k_1, k_2, \ldots, k_M$. Each stack contains $L$ layers. For each scale $m \in 1, 2, \ldots, M$ and layer $l \in 1, 2, \ldots, L$, we have:

$$Y_i^{local,m,l} = \text{ReLU}\left(\sum_{j=1}^{n} W_{m,l}[i-j] \cdot y_j^{local,m,l-1} + b_{m,l}\right), \tag{5}$$

where $W_{m,l}$ and $b_{m,l}$ are the weight matrix and bias term for the $m$-th scale and $l$-th layer 1D convolution, respectively, and ReLU is the rectified linear unit activation function. For the first layer ($l = 1$), we use the concatenated embeddings $x_j$ as the input.

After applying the 1D convolution layers, we apply a pooling layer to each scale to reduce the dimensionality:

$$Y_i^{pooled,m} = \text{Pooling}\left(Y_i^{local,m,L}\right). \tag{6}$$

We then concatenate the pooled local features from all scales:

$$Y_i^{pooled} = Y_i^{pooled,1}, Y_i^{pooled,2}, \ldots, Y_i^{pooled,M}. \tag{7}$$

The specific hyperparameter settings are as follows: We utilize three different scales of convolution, with kernel sizes of 3, 5, and 7. Each scale of convolution employs 64 filters, and the convolutional depth is set to 2.

**Hybrid attention layer.**   The proposed hybrid attention layer combines the global and local attention mechanisms to obtain a more comprehensive representation of the input sequence. We integrate the global and local attention mechanisms by using a linear layer (fully

connected layer) followed by a ReLU activation function to combine their outputs. The linear layer automatically adjusts the contribution of these two attention mechanisms based on the input data, finding the appropriate balance in the final output.

Given the output of the global attention layer, $Y^{global} \in \mathbb{R}^{n \times d}$, and the output of the local attention layer, $Y^{local} \in \mathbb{R}^{n \times d}$, where $n$ is the sequence length and $d$ is the feature dimension, concatenate these two outputs along the feature axis:

$$Y^{concat} = \text{Concatenate}(Y^{global}, Y^{local}) \in \mathbb{R}^{n \times (2d)}. \tag{8}$$

Next, combine the outputs of these two attention mechanisms using a linear layer. The weight matrix $W \in \mathbb{R}^{(2d) \times d}$ and bias vector $b \in \mathbb{R}^{d}$ of the linear layer are parameters that need to be learned during the training process. The output of the linear layer can be computed using the following equation:

$$Y^{preactivation} = Y^{concat} W + b. \tag{9}$$

Now, apply the ReLU activation function to the output of the linear layer to introduce non-linearity and obtain the final hybrid attention output $Y \in \mathbb{R}^{n \times d}$:

$$Y = \text{ReLU}(Y^{preactivation}). \tag{10}$$

In this way, the linear layer followed by the ReLU activation function automatically learns how to adjust the contribution of the global and local attention outputs, finding the optimal balance in the final output.

## Supervised contrastive learning

In the classification layer, we adopt supervised CL [41] to enhance the model's ability to distinguish RPIs from non-RPIs. The objective of contrastive representation learning is to learn an embedding space that preserves the similarity between samples of the same class while increasing the dissimilarity between samples of different classes. In this study, we incorporate a supervised CL training objective into the cross-entropy loss, which forms the final training objective for the model.

For a classification task with $C$ classes, the batch of training samples is $N$; $x_i, y_i$ represents the sample in batch and the label of the sample, respectively; $\Phi(x)$ denotes an encoder whose output is the result of the l2 normalization of the final hidden layer; $N_{y_i}$ represents the total number of samples with the same label $y_i$ in the batch; $\theta > 0$ is an adjustable scalar temperature parameter that controls the separation of classes; $y_{i,c}$ denotes the label of $y_i$; $\tilde{y}_{i,c}$ denotes the model output for the probability of the $ith$ example belonging to the class $c$; $\beta$ is a scalar weighted hyperparameter that is tuned for each downstream task and setting. The SCL loss is given in the following:

$$\mathcal{L}_{SCL} = \sum_{i=1}^{N} -\frac{1}{N_{y_i} - 1} \sum_{j=1}^{N} \mathbb{1}_{i \neq j} \mathbb{1}_{y_i = y_j} f(\cdot),$$

$$where \quad f(\cdot) = log \frac{exp(\Phi(x_i) \cdot \Phi(x_j)/\theta)}{\sum_{k=i}^{N} \mathbb{1}_{i \neq k} exp(\Phi(x_i) \cdot \Phi(x_k)/\theta)}. \tag{11}$$

The CE loss is given below:

$$\mathcal{L}_{\mathcal{CE}} = -\frac{1}{N}\sum_{i=1}^{N}\sum_{c=1}^{C} y_{i,c} \cdot \log \tilde{y}_{i,c} \ . \tag{12}$$

The overall loss is a weighted average of SCL and CE loss, as follows:

$$\mathcal{L} = (1 - \beta)\mathcal{L}_{\mathcal{CE}} + \beta\mathcal{L}_{\mathcal{SCL}}. \tag{13}$$

## Datasets

We evaluate the proposed method on five widely used datasets, NPInter2.0, RPI7317, RPI1807, RPI2241 and RPI369. Table 2 gives an overview.

The NPInter2.0 [42] dataset contains 10412 experimentally validated ncRPIs involving 4636 ncRNAs and 449 proteins from six organisms, where the sequences of ncRNAs and proteins are from the NONCODE database [43] and the UniProt [44] database, respectively. The RPI7317 dataset is a collection of 7317 experimentally validated human ncRPIs selected from the NPInter3.0 [45] database, involving 1874 ncRNAs and 118 proteins. The RPI2241 dataset contains 2241 interactions, involving 838 ncRNAs and 2040 proteins, and the RPI369 dataset contains 369 interactions, involving 331 ncRNAs and 338 proteins. The RPI1807 dataset contains 1807 interaction pairs involving 1078 RNAs and 3131 proteins. RPI2241, RPI369, RPI1807 data are extracted from RNA-protein binding complexes in the RNA-protein interaction database PRIDB [46] or PDB. The three datasets are constructed following the minimum atomic distance criterion, i.e., the distance between a protein atom and an RNA atom is less than a specified distance threshold, then the protein and RNA are considered as an interacting pair.

We construct an equal number of negative samples by randomly sampling RNA and protein pairs from the same dataset. However, to ensure the quality of the negative samples, we applied an identity threshold to filter out pairs that are similar to the interacting pairs. Specifically, for each sampled RNA and protein pair ($R_s$-$P_s$), if there exists an interacting pair ($R_i$-$P_i$) where $R_s$ and $R_i$ share more than 80% sequence identity, and $P_s$ and $P_i$ share more than 40% sequence identity, the sampled $R_s$-$P_s$ pair is discarded [13]. To cluster RNA and protein sequences, we used the CD-HIT 4.6.8 program [47], setting identity thresholds at 0.4 and 0.8. By this means, we aim to reduce false positives in the negative sample set and ensure the reliability of the evaluation results.

**Table 2. Datasets overview.**

| Dataset | Interaction Pairs | RNAs | Proteins |
|---|---|---|---|
| NPInter2.0 | 10412 | 4636 | 449 |
| NPI7317 | 7317 | 1874 | 118 |
| NPI2241 | 2241 | 838 | 2040 |
| RPI1807 | 1807 | 1078 | 3131 |
| RPI369 | 369 | 332 | 338 |

## Experiments

We evaluated the performance of our proposed RPIPLM method through rigorous five-fold cross-validation and compared it with the current state-of-the-art approaches. Our RPIPLM method is implemented using the Python programming language, leveraging PyTorch and Transformer packages, and executed on a computing unit consisting of 8 V100 GPUs.

### Metrics

In this study, we measure the performance of the models using accuracy, sensitivity (recall), and Matthews correlation coefficient (MCC), which are defined as follows:

$$Accuracy = \frac{TP + TN}{TP + TN + FP + FN}, \tag{14}$$

$$Recall = Sensitivity = \frac{TP}{TP + FN}, \tag{15}$$

$$MCC = \frac{TP \cdot TN - FP \cdot FN}{\sqrt{(TN + FN)(TN + FP)(TP + FN)(TP + FP)}}, \tag{16}$$

where TP, TN, FP, and FN are the number of true positives, true negatives, false positives, and false negatives, respectively. In addition, receiver operating characteristic curve (ROC) and AUC were also used for performance evaluation.

### Evaluation of RPIPLM

In this experiment, we thoroughly investigate the impact of protein pre-training models on the performance of RPIPLM by configuring 5 different pre-training models, including four pure sequence models (ProtBert-ref100, ProtBert-BFD, ProtBert-BFD , ProtElectra) and OntoProtein, a model fused with GO information. These protein pre-training models are combined with RNABERT to construct five RPIPLM models, namely RPIPLM-ProtBert-ref100, RPIPLM-ProtBert-BFD, RPIPLM-ProtAlbert, RPIPLM-ProtElectra, and RPIPLM-OntoProtein. To simplify the experiment, we concatenate the RNA and protein embeddings as $(r, p, |r - p|)$. We choose the NPInter2.0 dataset and RPI369 dataset for evaluation, which are the largest and smallest datasets among the five, respectively. We report the accuracy and MCC of all five models on these two datasets to comprehensively compare their performance.

We aim to investigate and provide insights into three fundamental questions regarding RPI prediction:

1. Can the incorporation of additional information, such as gene ontology (GO) knowledge, into protein pre-training models enhance the performance of RPI prediction?
2. Is it feasible to improve RPI prediction accuracy by expanding the corpus of pre-trained protein models?
3. Which approach is more effective in enhancing RPI prediction accuracy: expanding the pre-training corpus or incorporating GO information into the pre-training process?

To answer these questions, we conduct an in-depth evaluation of RPIPLM models that incorporate different protein pre-training models. Furthermore, we will analyze and compare the results to provide a comprehensive understanding of the impact of these pre-training models on RPI prediction accuracy.

**Results.** Firstly, we compared the performance of RPIPLM-ProtBert-ref100 and RPIPLM-OntoProtein to address whether incorporating protein function information through Onto-Protein improves RPI prediction. As observed from the comparison results presented in Fig 2, RPIPLM-OntoProtein achieved slightly better performance than RPIPLM-ProtBert-ref100. This implies that embedding protein function information indeed enhances the RPI prediction performance.

Secondly, to address the question of whether increasing the corpus of pre-trained protein models can improve RPI prediction, we compared the performance of RPIPLM-ProtBert-BFD and RPIPLM-ProtBert-ref100. Although these two models share the same architecture and training method, the former is pre-trained on the BFD dataset with 2.212 billion proteins, while the latter is pre-trained on UniRef100 with 261 million proteins. The results showed that RPIPLM-ProtBert-BFD achieved better performance, with 0.9% higher accuracy and 1.7% higher MCC on NPInter2.0 compared to RPIPLM-ProtBert-ref100. Similar results were observed on the RPI369 dataset.

Finally, we analyzed the results of the comparison in Fig 2 and found that RPIPLM-ProtBert-BFD outperformed RPIPLM-OntoProtein on both datasets. This suggests that

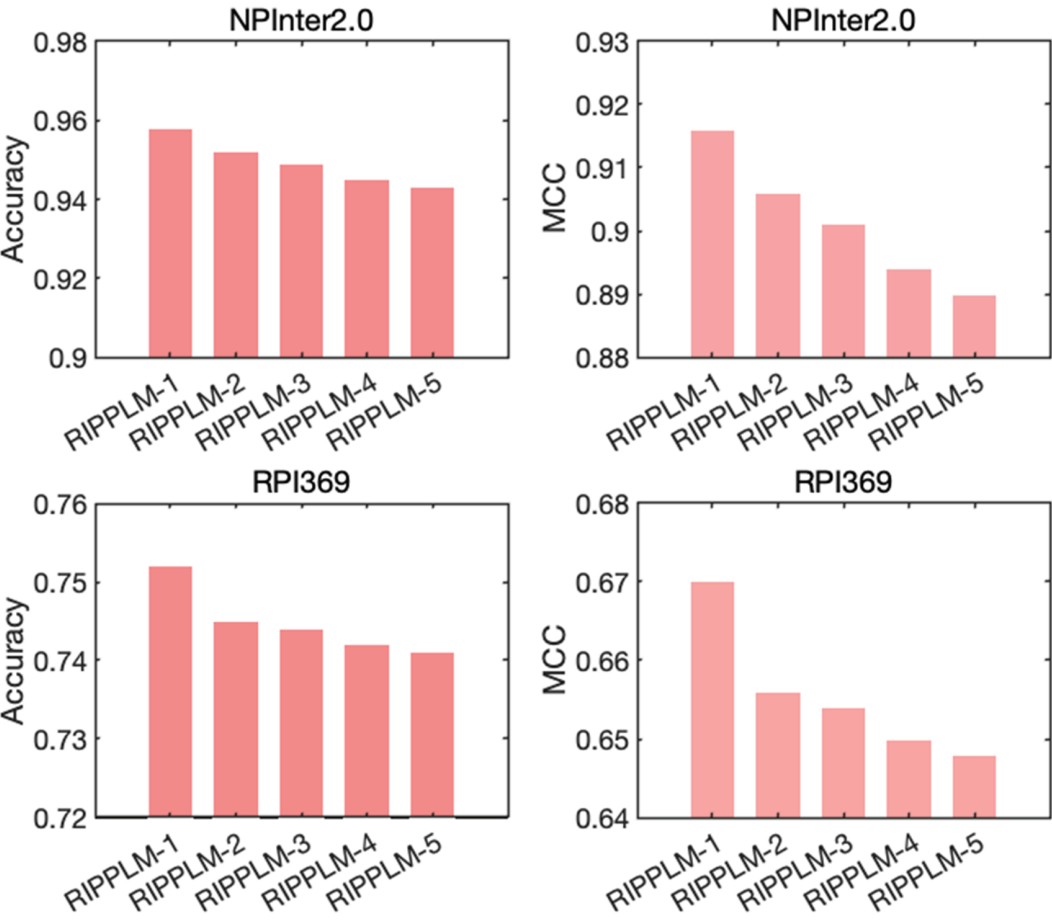

**Fig 2. Accuracy and MCC of 5 models on NPInter2.0 dataset and the RPI369 dataset.** RPIPLM-1 is RPIPLM-ProtBert-BFD, RPIPLM-2 is RPIPLM-OntoProtein, RPIPLM-3 is RPIPLM-ProtBert-ref100, RPIPLM-4 is RPIPLM-ProtElectra, RPIPLM-5 is RPIPLM-ProtAlbert.

expanding the protein sequence corpus provides more information than fusing GO knowledge, provided that computational resources are sufficiently available.

These findings not only answer the three questions posed in this experiment but also provide insightful conclusions. Additionally, since RPIPLM-ProtBert-BFD exhibited the best predictive performance, we adopt it as our model of choice for subsequent experiments, and refer to it simply as RPIPLM.

## Evaluation of the concatenation mode

In this experiment, our aim is to investigate the effectiveness of various modes of concatenating RNA and protein sequence embeddings in the two-tower model for RPI prediction [39], as described in Sect. To this end, we evaluate seven different concatenation modes on the NPInter2.0 and RPI2241 datasets, and report the results in Table 3.

We find that concatenation mode $(r, p, |r - p|)$ achieves the highest accuracy and best classification balance, suggesting that including both the raw embeddings and the differential embeddings (modules or element-wise product) can provide complementary information for RPI prediction. In contrast, concatenation modes $(|r - p|)$ and $(r*p)$, which aim to highlight differences between the RNA and protein embeddings, show lower performance compared to the direct concatenation mode $(r,p)$.

Overall, our results demonstrate that the choice of concatenation mode can have a significant impact on the performance of the two-tower model for RPI prediction, and highlight the importance of considering both raw and differential embeddings in the fusion process.

## Comparison with baseline methods

We compare RPIPLM performance against several state-of-the-art baseline methods, namely RPI-SE [19], IPMiner [14], RPITER [13], RPISeq-RF [9], NPI-GNN [22]. These baselines were chosen as they represent a range of methods that use different types of information for prediction, such as sequence information, network information, and structure information. By comparing RPIPLM against these baselines, we aim is to explore whether models that exploit general knowledge obtained from massive unlabeled sequence data can compete with methods that utilize complex information.

In Fig 3, we present the results of our RPIPLM model as compared to the baseline methods on five different datasets. The experimental results reveal that our proposed method outperforms the competing methods on all five datasets, with the highest classification accuracy and balance. Specifically, on the RIP369 dataset, our RPIPLM model exhibits a remarkable improvement of approximately 5% in accuracy, thus demonstrating its ability to effectively transfer extensive and comprehensive knowledge learned from large amounts of unlabeled

**Table 3. Test results of 7 concatenation modes.**

| Concatenation | NPInter2.0 | | RIP2241 | |
|---|---|---|---|---|
| | Acc | MCC | Acc | MCC |
| $(r,p)$ | 0.946 | 0.901 | 0.841 | 0.732 |
| $(|r - p|)$ | 0.934 | 0.882 | 0.834 | 0.675 |
| $(r*p)$ | 0.938 | 0.891 | 0.829 | 0.649 |
| $(|r - p|, r*p)$ | 0.945 | 0.902 | 0.838 | 0.689 |
| $(r,p,r*p)$ | 0.949 | 0.904 | 0.849 | 0.739 |
| $(r, p, |r - p|)$ | **0.958** | **0.919** | **0.865** | **0.748** |
| $(r, p, |r - p|, r*p)$ | 0.953 | 0.912 | 0.850 | 0.746 |

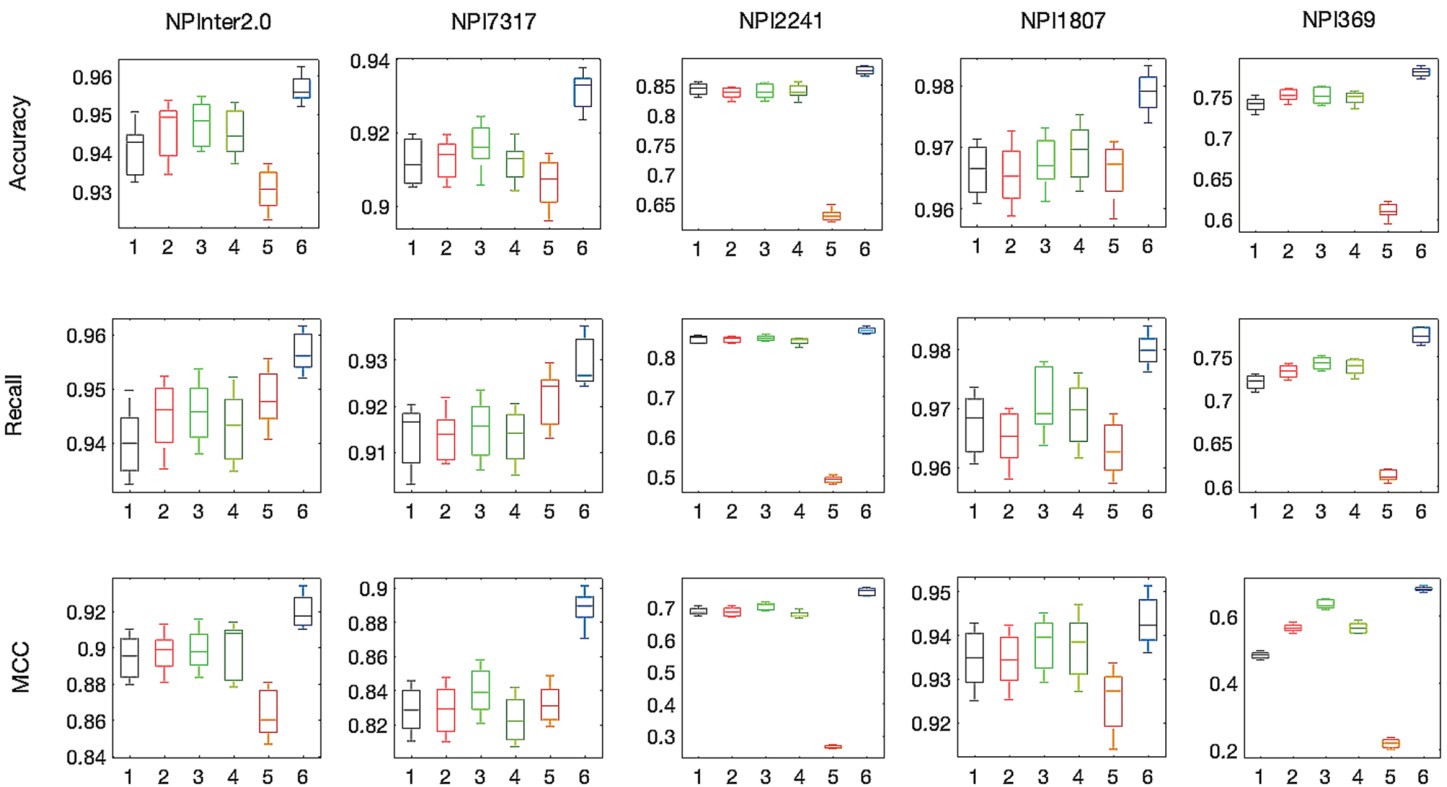

**Fig 3. Compare the accuracy, recall and ACC of RIPPLM and five baseline methods.** We use different colors and numbers to refer to each method: 1: RPI-SE; 2: IPMiner; 3: RPITER; 4: RIPSeq-RF; 5: NPI-GNN; 6: RPIPLM.

data using advanced self-supervised learning algorithms to labeled data learning. This capability highlights the potential of our model to scale-up the task of model learning on small datasets by leveraging the benefits of knowledge transfer.

In analyzing other methods, it is evident that IPMiner and RPITER, both complex ensemble models, have achieved promising performance across multiple datasets. Further scrutiny of these methods reveals that they employ a multi-tower structure and utilize both model ensembling and feature engineering. IPMiner, for instance, trains an integrated classifier by combining multi-layer embedded features with K-mer features, while RPITER uses RNA and protein sequence and structural data to train and integrate four deep networks. Comparatively, RPIPLM achieves higher accuracy predictions with a simpler and more intuitive structure, and a single source of information. This highlights the immense potential of pre-training in predicting RIPs, while affirming that the wealth of information obtained from massive unlabeled data through self-supervised learning can compensate for, and even surpass, the fusion of multiple-source information from labeled samples.

We also present a comprehensive comparison of RPIPLM with three selected baseline methods, in terms of ROC and AUC, as shown in Fig 4. Our experimental results demonstrate that RPIPLM achieves significantly better AUC values than current state-of-the-art methods on all datasets, including NPInter2.0, RPI7317, RPI2241, and RPI369. This observation suggests that RPIPLM has superior discriminative power and is more effective in distinguishing interactions from non-interactions. Moreover, we observe that the performance gap

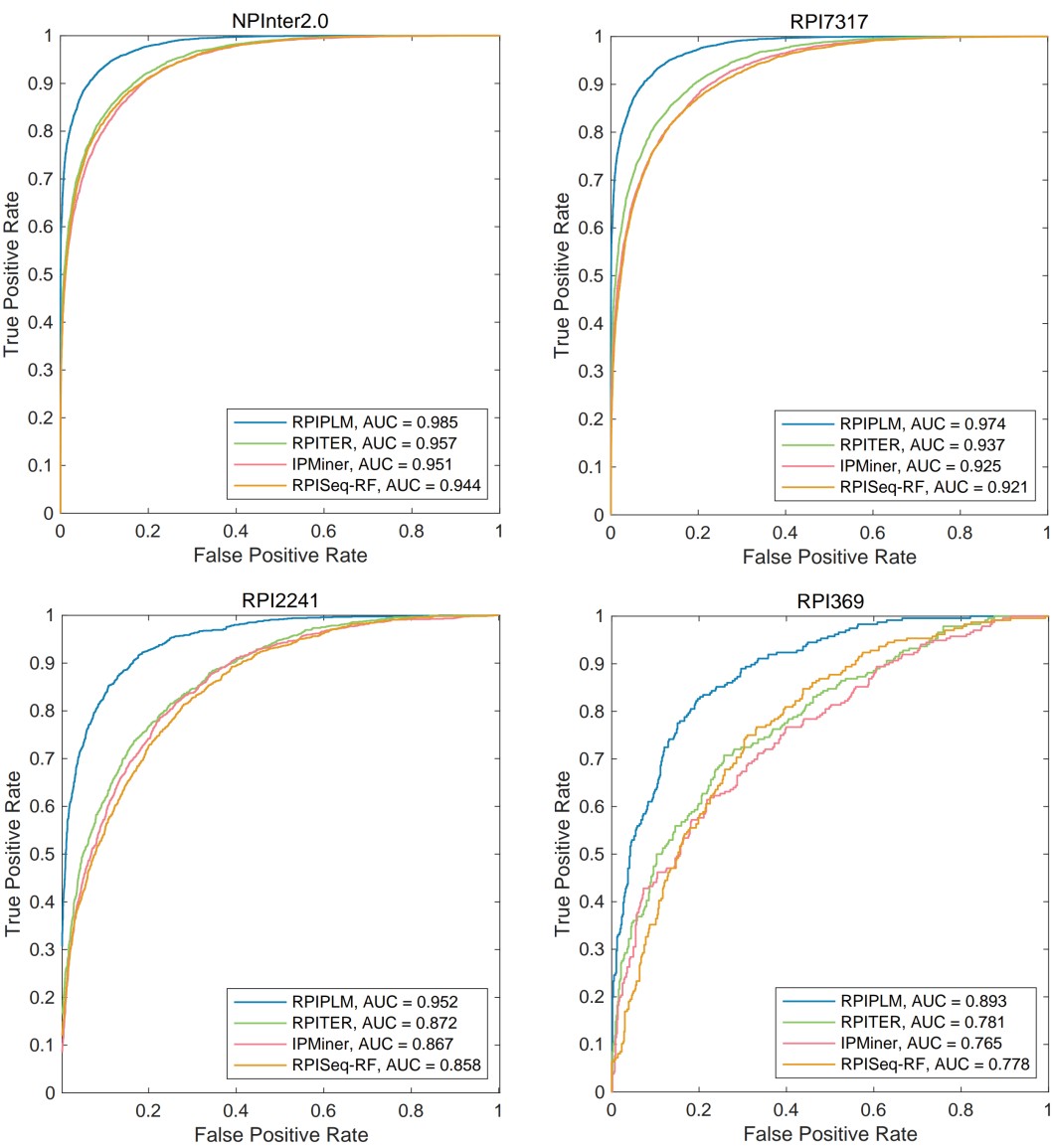

**Fig 4. Comparing ROC and AUC of RPIPLM and baseline methods.**

between RPIPLM and other methods gradually widens as the dataset size decreases. Specifically, on the RPI369 dataset with the least training data, RPIPLM outperforms other methods by more than 10% in terms of AUC, indicating that the model has a deeper understanding of the underlying patterns and relationships in the data. Overall, our findings underscore the effectiveness of RPIPLM and its potential applications in the field of biology laboratory.

## Performance on an independent dataset

We set up an independent dataset evaluation experiment, a more challenging experiment, in which we further evaluate the potential of the proposed method to predict novel RNA-protein interactions in practical applications. This experiment uses two independent and non-overlapping datasets, RPI369 and NPInter2.0, where the RPI369 dataset is used to train the

model and the NPInter v2.0 dataset is used for validation. There are 10,412 interaction pairs in the NPInter2.0 dataset, which are divided into 6: Homo sapiens, Caenorhabditis elegans, Mus musculus, Drosophila melanogaster, Saccharomyces cerevisiae, and Escherichia coli.

We present the evaluation results of a fine-tuned RPIPLM model on a set of six distinct organisms, as outlined in Table 4. The results demonstrate that the model, trained with only a limited number of samples, achieved an impressive accuracy of 0.9427 on a dataset of 10,412 previously unseen samples, accurately identifying interactions for each of the organisms with high precision. Notably, our analysis of the RIP369 dataset revealed a mere three interacting pairs of Caenorhabditis elegans organisms. The RPIPLM model identified 34 out of 36 air test samples for this organism, underscoring the model's exceptional performance in the task of interaction prediction. These results affirm that the RPIPLM model inherits powerful few-shot learning capabilities from its pre-trained model.

## Case study

In this study, we present case studies on two essential proteins, namely FUS from *Homo sapiens* and hnRNP-A2/B1 from *Mus musculus*. FUS is a multifunctional protein that plays a pivotal role in various cellular processes, including neuronal development and function, DNA damage repair, cell cycle regulation, and transcriptional regulation. Mutations in the FUS gene have been linked to numerous neurological disorders such as amyotrophic lateral sclerosis (ALS) and frontotemporal dementia (FTD) in humans. On the other hand, hnRNP-A2/B1 is expressed in many tissues in mice and plays a crucial role in fundamental biological processes such as alternative splicing, mRNA stability and translation, cell proliferation, differentiation and apoptosis. The use of RPI in mouse organisms facilitates the study of human diseases and genetic conditions, as well as the evaluation of the effects of drugs and other treatments.

We extract RNA from the datasets of *Homo sapiens* and *Mus musculus* included in NPInter2.0 and pair them with FUS and hnRNP-A2/B1 proteins, respectively. After removing the interacting pairs recorded in NPInter2.0, we use the trained RPIPLM model to predict potential interactions between unvalidated RNA-protein pairs. The predicted RNA-protein pairs are then ranked based on their interaction scores generated by the RPIPLM model, and we validate the top 10 pairs by checking whether they are documented in the updated versions of NPInter2.0, namely NPInter3.0 and NPInter4.0, which contain RPIs not included in the earlier version.

The experimental results are shown in Table 5, in which 4 RNAs interacting with FUS are verified, involving NONHSAT001945.2, NONHSAT067512.2, NONHSAT031447.2, NONHSAT115417.2; 3 RNAs interacting with hnRNP-A2/B1 are verified, involving NONHSAG015202.2, NONHSAG000479.3, NO-NHSAG020111.2. This validation procedure

**Table 4. Independent test on the NPInter2.0 dataset with the model trained on RPI369.**

| Organism | Total RNA-Protein interaction pairs | Correctly predicted | Accuracy |
|---|---|---|---|
| Caenorhabditis elegans | 36 | 34 | 0.9444 |
| Drosophila melanogaster | 91 | 85 | 0.9340 |
| Escherichia coli | 202 | 175 | 0.8663 |
| Saccharomyces cerevisiae | 910 | 887 | 0.9747 |
| Mus musculus | 2198 | 2107 | 0.9560 |
| Homo sapiens | 6975 | 6528 | 0.9359 |
| Total | 10412 | 9816 | 0.9427 |

**Table 5. RIPs predicted by RIPPLM and verified in the latest database.**

| RNA | Protein | Evidence | Organism |
|---|---|---|---|
| NONHSAT001945.2 | FUS | NPInter 4.0 | Homo sapiens |
| NONHSAT067512.2 | FUS | NPInter 4.0 | Homo sapiens |
| NONHSAT115417.2 | FUS | NPInter 4.0 | Homo sapiens |
| NONHSAT031447.2 | FUS | NPInter 4.0 | Homo sapiens |
| NONHSAG000479.3 | hnRNP-A2/B1 | NPInter 3.0 | Mus musculus |
| NONHSAG020111.2 | hnRNP-A2/B1 | NPInter 3.0 | Mus musculus |
| NONHSAG015202.2 | hnRNP-A2/B1 | NPInter 3.0 | Mus musculus |

provides evidence of the predicted interactions between RNA and proteins, highlighting the potential of the RPIPLM model in predicting RNA-protein interactions.

## Ablation study and performance impact

We conduct a series of ablation studies to investigate the contributions of various components of RPIPLM. These studies focus on two main aspects: the roles of the global and local attention modules, and the contributions of pre-training and contrastive learning (CL).

To investigate the contributions of the global attention, local attention, and their combination in RPIPLM, we conduct a series of ablation experiments on two datasets: NPInter2.0 and RPI2241. Table 6 summarizes the AUC and MCC scores under various configurations.

The experimental results demonstrate that both the global and local attention mechanisms contribute to enhancing the model's predictive capabilities to varying degrees. Specifically, compared to the full model, removing global attention leads to a 0.8% drop in AUC and 1.0% drop in MCC on NPInter2.0. The impact is even more pronounced on RPI2241, where AUC and MCC drop by 1.8% and 2.5%, respectively. Removing local attention has a similarly negative effect, with a 0.6% AUC drop and 1.7% MCC drop on RPI2241. When both modules are removed, performance degrades significantly: up to 3.4% lower MCC and 3.4% lower AUC compared to the full model. These results highlight the complementary roles of local and global attention mechanisms in capturing biologically meaningful RNA-protein interactions.

For the second investigation, we ablate the model into four distinct types and analyze their performance to investigate the impact of CL and pre-training on the performance of the RPIPLM model. The four types are as follows:

- No pre-training and no CL, denoted as No Pre & No CL;
- Pre-training without CL, denoted as Pre & No CL;
- No pre-training with CL, denoted as No Pre & CL;
- Full model with both pre-training and CL, denoted as Full.

**Table 6. Impact of global/local attention modules on model performance.**

| Ablation | NPInter2.0 | | RIP2241 | |
|---|---|---|---|---|
| | **AUC** | **MCC** | **AUC** | **MCC** |
| No global attention | 0.977 | 0.910 | 0.935 | 0.729 |
| No local attention | 0.979 | 0.912 | 0.928 | 0.701 |
| without both | 0.973 | 0.901 | 0.920 | 0.669 |
| with both | **0.985** | **0.919** | **0.952** | **0.748** |

By comparing the performance of the four types of the model, we aim to understand the relative contribution of pre-training and CL to the overall performance of RPIPLM. This analysis can provide insights into the role of pre-training and CL in facilitating the prediction of RNA-protein interactions and can aid in developing more effective models for this task.

We apply the four types of the RPIPLM model to each of the five datasets and evaluate their performance by plotting receiver operating characteristic (ROC) curves and reporting area under the curve (AUC) values for comparison. Our results, as shown in Fig 5, indicate that incorporating pre-training or CL can improve the performance of RPIPLM in recognizing RPIs, with the model that combines both pre-training and CL achieving the greatest improvement.

To further investigate the relative contributions of pre-training and CL to RPI prediction, we compare the performance of the No Pre & CL and Pre & No CL models. We observe that No Pre & CL performs slightly better than Pre & No CL on the RIP2241 and RIP369 datasets. These results suggest that CL may have a greater impact on RPI prediction when labeled data is scarce, whereas pre-training is more effective when sufficient labeled data is available.

Analyzing the technical characteristics of pre-training and CL, we can gain insights into the improvements achieved by the RPIPLM model in predicting RNA-protein interactions.

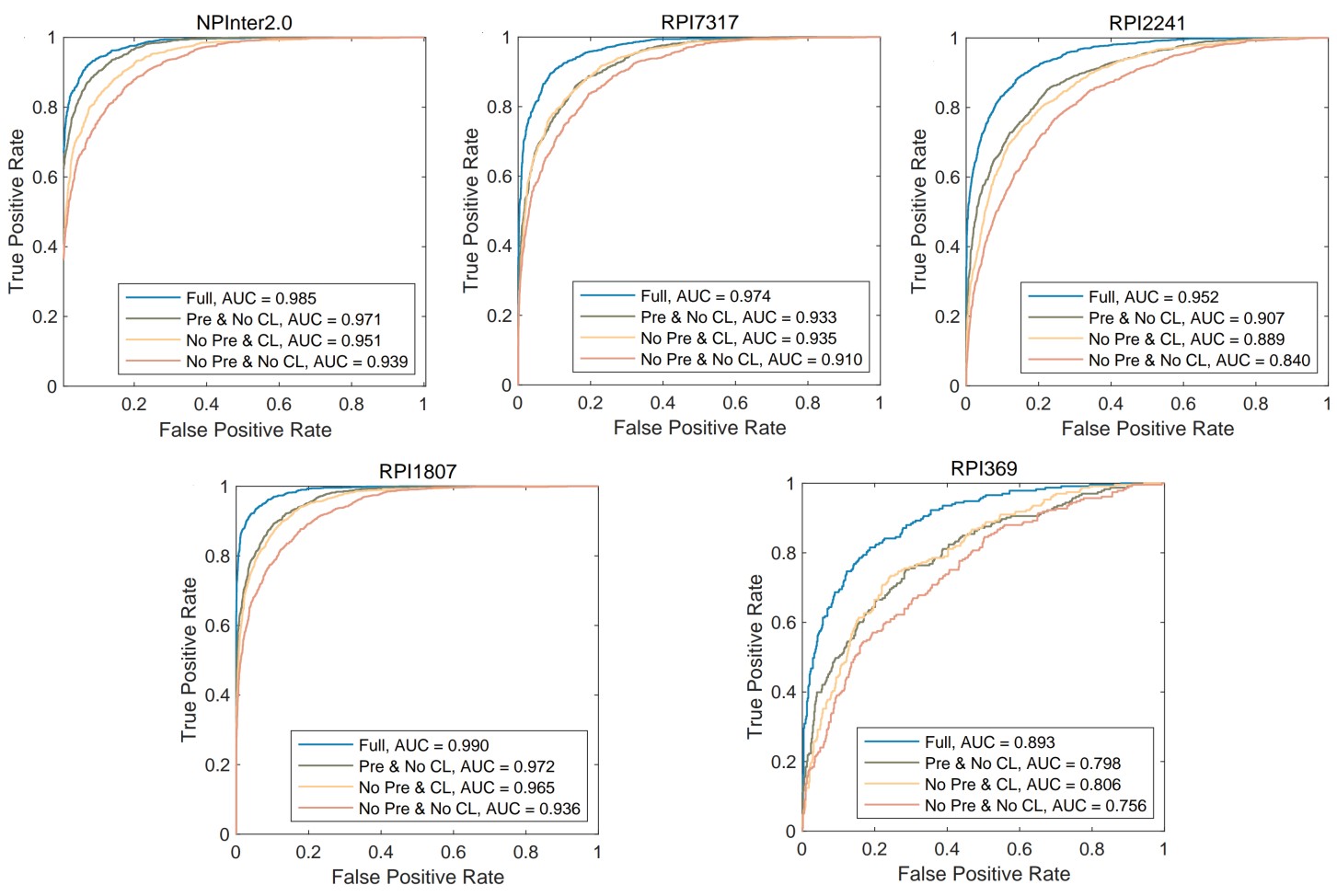

**Fig 5. Results of model ablation experiments on 5 datasets.**

Specifically, pre-training and CL enhance RIPs prediction from two distinct perspectives. Pre-training optimizes the expression of embeddings by translating new information into the learned model, while CL emphasizes the separation of embedding vectors in space to improve class distinguishability.

Since these two approaches are non-conflicting and non-redundant, their combination in the RPIPLM model produces a synergistic effect, which leads to improved performance in predicting RNA-protein interactions. This highlights the importance of considering multiple technical approaches in developing effective models for complex tasks such as RNA-protein interaction prediction. Our findings offer new insights into the technical underpinnings of pre-training and CL and their contributions to the development of the RPIPLM model.

## Model interpretability

This section employs visualization techniques to provide intuitive insights into how RPIPLM achieves state-of-the-art performance across all evaluations. Specifically, we use t-distributed stochastic neighbor embedding (T-SNE) to map the output of the last layer of RPIPLM in a 2D space. This allows us to visualize the distribution of embeddings and the distinction boundary between interacting and non-interacting pairs.

We compare the output of the last layer of RPIPLM with that of RPITER, the baseline method that performs best overall. This comparison allows us to better interpret the performance of RPIPLM and its technical features. Additionally, we show the visualization results of the ablated models discussed in Sect (see Fig 6), providing further insights into the roles of pre-training and CL in enhancing the performance of RPIPLM.

Our analysis reveals that RPIPLM produces feature distributions that are more class-discriminative than the previous best method. Specifically, RPIPLM-processed interacting and non-interacting pairs exhibit the least overlap in space, the largest inter-class distance, and a higher degree of intra-class aggregation, indicating more effective separation of the two classes. Furthermore, the distributions of the two classes of data visualized by RPIPLM appear to be flatter, indicating smaller correlations between the sample characteristics and higher expression of key distinguishing information.

Our results also show that the embedding features of the samples extracted by the RPIPLM model configured with CL are more spatially cohesive, further supporting the effectiveness of CL in reducing the distance between the classes. By comparing the visualization results

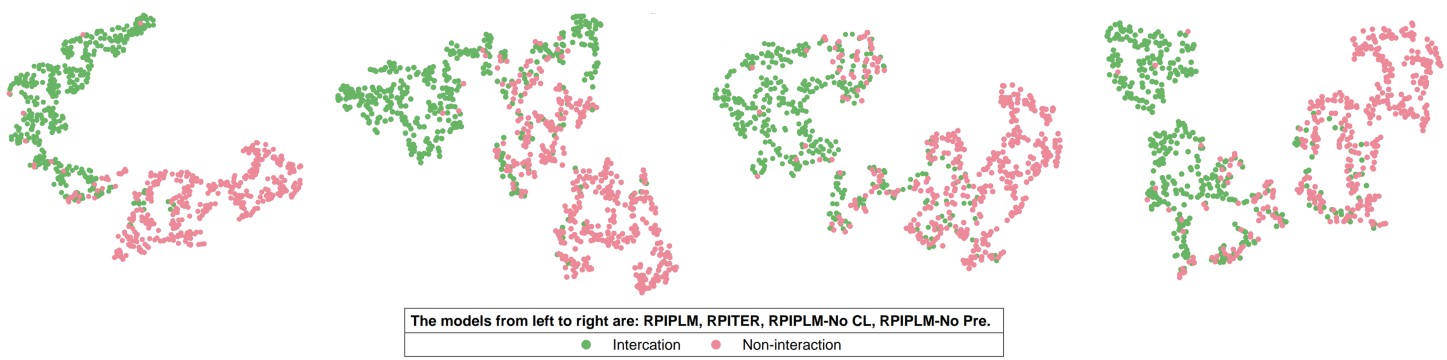

The models from left to right are: RPIPLM, RPITER, RPIPLM-No CL, RPIPLM-No Pre.

● Intercation ● Non-interaction

**Fig 6. T-SNE visualization of the last layer output embeddings of the RIPPLM series models and the RPITER model.** Data:RPI369.

of the two ablation versions of RPIPLM, we gain deeper insights into the technical characteristics and contributions of pre-training and CL to the model's performance in predicting RNA-protein interactions.

These findings offer new insights into the technical underpinnings and performance of RPIPLM, highlighting the importance of incorporating both pre-training and CL to achieve effective separation of interacting and non-interacting pairs.

## Conclusion

The crux of our research endeavor rests upon the unveiling of a novel computational framework that aspires to revolutionize the realm of ncRNA-protein interaction prediction. The heart of this innovative framework lies in its employment of a twin-tower structure, which synergistically amalgamates RNA and protein pre-trained models, thereby endowing us with the capability to transfer knowledge from the vast domain of unsupervised learning to the RPI prediction task. Notably, our twin-tower pre-training framework stands out from the crowd by integrating a supervised contrastive learning task, which intentionally distances interacting and non-interacting pairs in space, thereby enhancing its fine-tuning potential.

Pleasingly, our proposed method has yielded unparalleled performance and accuracy when compared with the current state-of-the-art methods for RPI prediction, a fact substantiated by our extensive experiments and demonstrations. By forging a new path in the field of ncRNA-protein interaction prediction, our research contributes to the advancement of computational tools, thus facilitating a deeper understanding of the intricate processes that underlie cellular dynamics, and ultimately propelling us towards the discovery of novel drugs and therapeutics.

Beyond performance, RPIPLM offers potential for further biological interpretability. Future work will focus on analyzing attention weights to identify whether the model implicitly attends to known or novel binding regions in RNA and proteins. This line of investigation may help bridge predictive modeling with biological insight, ultimately enabling interpretable and discovery-driven RNA–protein interaction analysis.

## Author contributions

**Data curation:** Yiwei Liu, Ting Bao.

**Investigation:** Peng Yin.

**Methodology:** Yiwei Liu, Shumin Wang, Yanbin Wang.

**Validation:** Yiwei Liu.

**Visualization:** Yiwei Liu, Ting Bao, Peng Yin.

**Writing – original draft:** Shumin Wang, Yanbin Wang.

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
