## [Decision Letter · Decision Letter 0]

18 Dec 2023

PONE-D-23-34195RPIPLM: prediction of ncRNA-protein interaction by post-training a dual-tower pretrained biological model with Supervised Contrastive LearningPLOS ONE

Dear Dr. Liu,

Thank you for submitting your manuscript to PLOS ONE. After careful consideration, we feel that it has merit and may be considered for publications after a thorough revision. Therefore, we invite you to submit a revised version of the manuscript that addresses the points raised during the review process. Please attend to the comments and questions of the reviewers particularly related to the experimental setup and results. More specifically, I would expect your response and if appropriate revision on Points #4-#7 of Reviewer 2 and the points listed under the "Weak Points/possible Improvements" of Reviewer 1.

We look forward to receiving your revised manuscript.

Kind regards,

M. Sohel Rahman, Ph.D.

Academic Editor

PLOS ONE

Journal Requirements:

"We gratefully acknowledge the financial support received from the following funding sources to conduct this research. The research was partially supported by the Defense Industrial Technology Development Program (Grant

JCKY2021906B002 and Grant JCKY2021602B002)."

"The authors received no specific funding for this work."

Reviewers' comments:

Reviewer's Responses to Questions

**Comments to the Author**

1. Is the manuscript technically sound, and do the data support the conclusions?

Reviewer #1: Yes

Reviewer #2: Yes

2. Has the statistical analysis been performed appropriately and rigorously? 

Reviewer #1: Yes

Reviewer #2: Yes

3. Have the authors made all data underlying the findings in their manuscript fully available?

Reviewer #1: Yes

Reviewer #2: Yes

4. Is the manuscript presented in an intelligible fashion and written in standard English?

Reviewer #1: Yes

Reviewer #2: Yes

5. Review Comments to the Author

Reviewer #1: Summary: The aforementioned paper introduces an innovative method called RPIPLM, designed to predict interactions between non-coding RNA (ncRNA) and proteins.

The method applied a pair of parallel BERT-based models to encode both the ncRNA and protein sequences simultaneously. Then these two encodings were concatenated to form a unified feature representation which is refined by an attention based method. After which a contrastive learning based model predicts whether the ncRNA-protein pair interacts. To train the BERT-based models, unlabelled ncRNA and protein sequences were used (they were pre trained so authors didn't need to train them). Also labelled data was used to train a number of parameters of the pipeline. The main claim of the authors is that using the BERT-models harnesses the power of huge amounts of unlabelled data that improve ncDNA-Protein interaction prediction, which is shown in the results section.

Strong Points:

The world renowned BERT model was used in an intricate way.

Vast quantities of Unlabelled data is available for protein or RNA sequences. The RPIPLM model’s attempt to utilise unlabelled data is praiseworthy.

A case study was conducted to demonstrate the model's performance in real-world scenarios.

For protein encoding, multiple BERT models were tested. And the employment of OntoProtein was very thoughtful.

Instead of using hard coded thresholds, ROC curve was shown for comparison, which is a plus point.

Results generated by using various concatenation methods were shown (Table 3).

Weak Points/possible Improvements:

In Figure 1, the protein sequence only has A C G U. But generally there are 20 different amino acid residues. This may confuse some readers. Also the protein sequence looks identical to RNA sequence. Which may falsely make some readers think sequences can be identical. (Yes, the ‘...’, means anything can be there but many readers may skip this). Also in the FIgure where MLM is shown the sequences do not match the input.

In section 2.2, some concatenation methods are stated and what each concatenation method captures is given, such as , “This method provides a straightforward

approach to concatenate RNA and protein features, but may not capture more complex interactions between RNA and protein.” - what is the basis of the underlined states? Similarly for other concatenation methods such a statement is given. If there were references it would be better.

In section 2.3.2, kernel size of convolutional layers, filter count and depth is given which were fixed during tests. It would be better if multiple tests were done with different values for these hyperparameters.

In section 3, it is assumed that, if ncRNA “Rs and Ri share more than 80% sequence identity” or proteins “Ps and Pi share more than 40%” sequence identity” than if pair Rs and Ps interact then Ri and Pi interact. This way the dataset was filtered. What is the basis of choosing these percentages (80% and 40%).

In section 4.7, we see there were tests with No pre-training and no CL. It is unclear how such this model was trained and tested.

RNABERT used RNA structural information in its pipeline. For protein encoding pipeline addition of such options can be great, as huge protein structural databases are available.

There are great tools such as Hmmer or HH-Suite that can create MSA from or do profiling for protein sequences. What happens when results of these tools are used to again tune the MLM encoded results? Such tasks may improve performance.

I think once the weaknesses/questions are addressed this could be a decent contribution.

Reviewer #2: 1. It would be great if the authors included line numbers.

2. In the second paragraph, some of the examples were repeated in the sentence "These interactions are involved in a range of biological processes, including disease development, cellular signaling, and gene expression regulation, among others.". Either provide new examples or remove these lines.

3. The introduction section is over-sized. Too much focus on previous work. It is better to reduce the number of examples and their contributions. Include one/two from machine learning and two/three from deep learning with the most ground breaking contributions at their time.

4. There are other model for protein feature extraction like ESM and ProteinBERT. Why have those not been tested?

5. To calculate |r-p| and r*p, both r and p need to be of same size. However, the pretrained models don't give output of same size. How did the authors convert them to same size? Dense layer or 1D conv? Need to mention that.

6. The global and local attention methods were not described. From the figure it seems, global attention is scaled dot product attention by Vaswami and local attention is regular dot product attention. Should detail on these and refer to the authors of the original work.

7. Please highlight the best scores in Table 3.

6. PLOS authors have the option to publish the peer review history of their article (what does this mean?). If published, this will include your full peer review and any attached files.

Reviewer #1: No

Reviewer #2: **Yes: **Gourab Saha

---

## [Author Response · Author response to Decision Letter 1]

25 Mar 2024

We express our gratitude to the esteemed reviewer for providing constructive feedback and valuable suggestions regarding the positive evaluation of our work. Furthermore, his/her insightful recommendations have significantly contributed to improving the quality and rigor of our manuscript. Please refer to our submitted PDF document "Response to Reviewers" for specific responses.

---

## [Decision Letter · Decision Letter 1]

18 Jun 2024

PONE-D-23-34195R1RPIPLM: prediction of ncRNA-protein interaction by post-training a dual-tower pretrained biological model with Supervised Contrastive LearningPLOS ONE

Dear Dr. Liu,

Thank you for submitting your manuscript to PLOS ONE. After careful consideration, we feel that it has merit but does not fully meet PLOS ONE’s publication criteria as it currently stands. Therefore, we invite you to submit a revised version of the manuscript that addresses the points raised during the review process.

Pleas note that while Reviewer 1 seems to be happy with your revision effort, Reviewer 2 is still unhappy and suggested a Major revision. While, I felt the recommendation of Reviewer 2 is a bit harsh, I do agree with his points, particularly, Points 3 and 6. Therefore, a further revision is in order or you may provide a clear rebuttal if you believe that the comments are unjustified.==============================

We look forward to receiving your revised manuscript.

Kind regards,

M. Sohel Rahman, Ph.D.

Academic Editor

PLOS ONE

Journal Requirements:

Reviewers' comments:

Reviewer's Responses to Questions

**Comments to the Author**

1. If the authors have adequately addressed your comments raised in a previous round of review and you feel that this manuscript is now acceptable for publication, you may indicate that here to bypass the “Comments to the Author” section, enter your conflict of interest statement in the “Confidential to Editor” section, and submit your "Accept" recommendation.

Reviewer #1: All comments have been addressed

Reviewer #2: (No Response)

2. Is the manuscript technically sound, and do the data support the conclusions?

Reviewer #1: Yes

Reviewer #2: Partly

3. Has the statistical analysis been performed appropriately and rigorously? 

Reviewer #1: Yes

Reviewer #2: Yes

4. Have the authors made all data underlying the findings in their manuscript fully available?

Reviewer #1: Yes

Reviewer #2: No

5. Is the manuscript presented in an intelligible fashion and written in standard English?

Reviewer #1: Yes

Reviewer #2: Yes

6. Review Comments to the Author

Reviewer #1: All the comments have been addressed properly. Relevant references were added and the manuscripts was properly updated. I am happy with the revised manuscript.

Reviewer #2: 1. Table-3 needs to show the best score per column in bold.

2. The authors need to show the applicability of the method on SOTA PLMs like ESM (https://github.com/facebookresearch/esm) and RNA LMs like RiNALMo (https://arxiv.org/pdf/2403.00043) to prove robustness of the idea.

3. One major issue with the method is the application of dot product attention and scaled dot product attention on the concatenated features. Attention is usually applied along the length of a sequence. However, this method applies it along the feature dimension. The article never explained the rationale behind it If it wasn't the case, the description is not well written to reflect that.

4. Table-6 in Ablation studies should indicate the best scores in bold. Additionally, authors might reconsider a different title for the section. Additionally, in the description, instead of rehashing what the table shows, authors may instead focus on the improvement ( in percentage or other measures).

5. Additionally, while the difference in performance may not be big as seen in the ablation studies, it would be interesting to see the effects of the various attention modules on the leaned representations. Authors may perform the TSNE plot experiments to show that. Also, if there are significant differences, an interesting study would be exploring how the attention modules impact the feature space.

7. PLOS authors have the option to publish the peer review history of their article (what does this mean?). If published, this will include your full peer review and any attached files.

Reviewer #1: **Yes: **Sabab Aosaf

Reviewer #2: No

---

## [Author Response · Author response to Decision Letter 2]

11 Apr 2025

We express our gratitude to the esteemed reviewer for providing constructive feedback and valuable suggestions regarding the positive evaluation of our work. Furthermore, his/her insightful recommendations have significantly contributed to improving the quality and rigor of our manuscript.

1. Reviewer #1

1.1. Comment 1

RC: All the comments have been addressed properly. Relevant references were added and the manuscripts was properly updated. I am happy with the revised manuscript.

AR: Thank you for your positive feedback and for your time and effort in reviewing our manuscript.

2. Reviewer #2

2.1. Comment 1

RC: Table-3 needs to show the best score per column in bold.

AR: Thank you for your suggestion. We have made the necessary revisions to Table 3 to show the best score per column in bold in the revised manuscript.

2.2. Comment 2

RC: The authors need to show the applicability of the method on SOTA PLMs like ESM(https://github.com/

facebookresearch/esm) and RNA LMs like RiNALMo (https://arxiv.org/pdf/2403.00043) to prove robustness of the idea.

AR: Thank you for your valuable suggestion. We agree that testing on state-of-the-art PLMs such as ESM and RiNALMo would further demonstrate the robustness and generalizability of our method. As these models have different input formats and computational demands, we plan to include them in our future work to extend RPIPLM’s applicability. In the current version, we have focused on integrating and evaluating a range of competitive PLMs (e.g., ProtBert, OntoProtein, RNABERT) to establish the foundational performance of our dual-tower framework.

2.3. Comment 3

RC: One major issue with the method is the application of dot product attention and scaled dot product attention on the concatenated features. Attention is usually applied along the length of a sequence. However, this method applies it along the feature dimension. The article never explained the rationale behind it If it wasn't the case, the description is not well written to reflect that.

AR: We appreciate the reviewer’s insightful comment. However, we believe there may have been a misunderstanding. In our method, the dot-product attention is applied along the sequence (token) dimension, not along the feature (embedding) dimension.

As described in Section Global and Local Attention Modules (page 7), we define our input $X = [x_1, x_2, \dots, x_n]$ as a sequence of embeddings, where $n$ denotes the sequence length (i.e., the number of tokens after concatenation of RNA and protein CLS or pooled embeddings). The scaled dot-product attention then computes attention weights between all pairs of tokens in this sequence. To avoid further confusion, we have revised the manuscript to explicitly state the dimension along which attention is applied, and we have clarified that the attention is not across feature channels.

2.4. Comment 4

RC: Table-6 in Ablation studies should indicate the best scores in bold. Additionally, authors might reconsider a different title for the section. Additionally, in the description, instead of rehashing what the table shows, authors may instead focus on the improvement ( in percentage or other measures).

AR: Thank you for your insightful observation.Thank you for the constructive suggestions regarding the ablation study section. We have updated Table 6 by highlighting the best-performing scores in bold for clearer comparison. Additionally, we have renamed the section to “Ablation Study and Performance Impact” to better reflect its content. In the revised description, we now focus on the relative performance changes, reporting the percentage improvements and degradations caused by the removal of key modules, rather than restating the table values.

2.5. Comment 5

RC: Additionally, while the difference in performance may not be big as seen in the ablation studies, it would be interesting to see the effects of the various attention modules on the leaned representations. Authors may perform the TSNE plot experiments to show that. Also, if there are significant differences, an interesting study would be exploring how the attention modules impact the feature space.

AR: We thank the reviewer for the valuable suggestion to explore the impact of attention modules on the learned representations. While we initially planned to use t-SNE visualizations, we agree that a quantitative summary better complements our existing evaluation style.

To this end, we conducted embedding-level ablation experiments under the same concatenation scheme ($(r, p, |r - p|)$), where we removed either global attention, local attention, or both. The classification results using the derived embeddings are reported in the following table.These results show that removing either attention module consistently degrades performance, with the most notable drop occurring when both are removed. This supports the view that the attention mechanisms help shape a more discriminative and informative feature space, even when the top-line metrics are not drastically different.

Configuration Accuracy (NPInter2.0) MCC (NPInter2.0) Accuracy (RPI2241) MCC (RPI2241)

No Global Attention 0.931 0.875 0.812 0.667

No Local Attention 0.922 0.868 0.801 0.643

No Global & Local Attention 0.903 0.847 0.785 0.620

---

## [Editor Report · Decision Letter 2]

6 May 2025

PONE-D-23-34195R2RPIPLM: prediction of ncRNA-protein interaction by post-training a dual-tower pretrained biological model with Supervised Contrastive LearningPLOS ONE

Dear Dr. Liu,

Thank you for submitting your manuscript to PLOS ONE. After careful consideration, we feel that it has merit but does not fully meet PLOS ONE’s publication criteria as it currently stands. Therefore, we invite you to submit a revised version of the manuscript that addresses the points raised during the review process.

We look forward to receiving your revised manuscript.

Kind regards,

M. Sohel Rahman, Ph.D.

Academic Editor

PLOS ONE

Journal Requirements:

Additional Editor Comments:

I have been in touch with the reviewer who had trouble submitting the report. So, I am communicating his comments for you (the authors) for (minor) revision.

1. Because the model is trained in a fully supervised manner, its final embedding layer is inherently optimized for class separation, so any t-SNE projection will directly mirror its classification accuracy. As a result, the only distinction between the RPIPLM and RPITER t-SNE visualizations arises from their differing accuracy ( and other metrics) scores, and the “new” plots offered are in fact identical to those already shown in Figure 3. Additionally, the 3rd and 4th plots in the tSNE plots are reflective of the final plot in Figure 5.

2. As for what the authors can do to show the interpretability of the model

They can use the attention wits to see which tokens the model is paying most attention to while classifying and if those tokens have patterns that might represent something biologically.

Biologically interpret whether the model’s attention weights are indicative of the binding sites, and if the weights can shed some light on that.

Note that authors are not restricted to these ideas and can explore any other ideas they might find a good fit.

3. All other comments have been addressed

---

## [Author Response · Author response to Decision Letter 3]

4 Jun 2025

We express our gratitude to the esteemed reviewer for providing constructive feedback and valuable suggestions regarding the positive evaluation of our work. Furthermore, his/her insightful recommendations have significantly contributed to improving the quality and rigor of our manuscript.

1. Comment 1

RC: Because the model is trained in a fully supervised manner, its final embedding layer is inherently optimized for class separation, so any t-SNE projection will directly mirror its classification accuracy. As a result, the only distinction between the RPIPLM and RPITER t-SNE visualizations arises from their differing accuracy ( and other metrics) scores, and the “new” plots offered are in fact identical to those already shown in Figure 3. Additionally, the 3rd and 4th plots in the tSNE plots are reflective of the final plot in Figure 5.

AR:

We thank the reviewer for the thoughtful comment and agree with the observation that “any t-SNE projection will directly mirror its classification accuracy.” Indeed, as our model is trained in a fully supervised manner, the embedding space is naturally optimized for class separation, and the resulting t-SNE plots primarily reflect this objective.

However, we would like to clarify two points:

1.We are not entirely certain about the reference to “the RPITER t-SNE visualizations”, as RPITER was not included in any t-SNE comparison in our manuscript. Furthermore, we did not introduce any new t-SNE plots in the revised version of the manuscript beyond what was already present in the original submission (Figure 6). If there has been a misunderstanding on our part, we would be grateful for clarification.

2.Regarding the comment that “the 3rd and 4th plots in the t-SNE plots are reflective of the final plot in Figure 5,” we believe this refers to a perceived overlap in visual information. We note that the last plot in Figure 5 shows the performance of RPIPLM on the RPI369 dataset, and the t-SNE example we included in Figure 6 is likewise drawn from RPI369. Therefore, the similarity is expected and intentional: the t-SNE plot serves as a geometric visualization corresponding to the same experimental condition.

2. Comment 2

RC: As for what the authors can do to show the interpretability of the model They can use the attention wits to see which tokens the model is paying most attention to while classifying and if those tokens have patterns that might represent something biologically. Biologically interpret whether the model’s attention weights are indicative of the binding sites, and if the weights can shed some light on that.

Note that authors are not restricted to these ideas and can explore any other ideas they might find a good fit.

AR:

We appreciate the reviewer’s insightful suggestion regarding the use of attention weights for biological interpretability. Indeed, analyzing which tokens receive high attention may help identify sequence regions associated with binding activity. While preliminary inspection of attention maps shows that the model occasionally highlights regions near known binding sites, a more rigorous biological validation would require curated site-level annotations, which we plan to incorporate in follow-up studies.

---

## [Editor Report · Decision Letter 3]

13 Jul 2025

RPIPLM: prediction of ncRNA-protein interaction by post-training a dual-tower pretrained biological model with Supervised Contrastive Learning

PONE-D-23-34195R3

Dear Dr. Liu,

We’re pleased to inform you that your manuscript has been judged scientifically suitable for publication and will be formally accepted for publication once it meets all outstanding technical requirements.

Kind regards,

M. Sohel Rahman, Ph.D.

Academic Editor

PLOS ONE

Additional Editor Comments (optional):

The reviewer (who had trouble accessing the system and hence the delay), has made the following comments to me:

1. As the authors mentioned that the overlap between the plots in figure-5 and 6 are intentional , it would be best if the tSNE plots were moved from the model interpretability section and moved to the result discussion section. 

2. Rest of the comments have been addressed.

I would request the authors to consider #1 above as a discretionary comment. 